# The Evolution and Growth Patterns of the Road Network in a Medium-Sized Developing City: A Historical Investigation of Changchun, China, from 1912 to 2017

**Shiguang Wang** [1,2,4,5], **Dexin Yu** [1,4,5], **Mei-Po Kwan** [2,3] , **Huxing Zhou** [1,4,5],* , **Yongxing Li** [6] **and Hongzhi Miao** [1]

1 School of Transportation, Jilin University, Changchun 130022, China; wsg_its@163.com or wangsg16@mails.jlu.edu.cn (S.W.); yudx@jlu.edu.cn (D.Y.); mouhz16@mails.jlu.edu.cn (H.M.)
2 Department of Geography and Geographic Information Science, University of Illinois at Urbana—Champaign, Urbana, IL 61801, USA; mkwan@illinois.edu
3 Department of Human Geography and Spatial Planning, Utrecht University, Utrecht 3508, The Netherlands
4 Jilin Research Center for Intelligent Transportation System, Changchun 130022, China
5 Jilin Province Key Laboratory of Road Traffic, Changchun 130022, China
6 School of Civil and Environmental Engineering, Nanyang Technological University, Singapore 639798, Singapore; yongxing.li@ntu.edu.sg
* Correspondence: zhouhx@jlu.edu.cn

**Abstract:** Understanding the evolution and growth patterns of urban road networks helps to design an efficient and sustainable transport network. The paper proposed a general study framework and analytical workflow based on network theory that could be applied to almost any city to analyze the temporal evolution of road networks. The main tasks follow three steps: vector road network drawing, topology graph generation, and measure classification. Considering data availability and the limitations of existing studies, we took Changchun, China, a middle-sized developing city that is seldom reported in existing studies, as the study area. The research results of Changchun (1912–2017) show the road networks sprawled and densified over time, and the evolution patterns depend on the historical periods and urban planning modes. The evolution of network scales exhibits significant correlation; the population in the city is well correlated with the total road length and car ownership. Each network index also presents specific rules. All road networks are small-world networks, and the arterial roads have been consistent over time; however, the core area changes within the adjacent range but is generally far from the old city. More importantly, we found the correlation between structure and function of the urban road networks in terms of the temporal evolution. However, the temporal evolution pattern shows the correlation varies over time or planning modes, which had not been reported

**Keywords:** urban road network; evolution pattern; network theory; medium-sized city; structure and function.

## 1. Introduction

Transportation is one of the most critical functions in cities. To design an efficient and sustainable transport network, urban and transport planners need to understand the evolution and growth patterns of urban road networks. The evolution and growth patterns are a kind of structural characteristic of a network. Network science provides powerful theoretical and practical tools to analyze the structure of complex networks. Many studies have used quantitative methods and techniques based on network

theory to analyze diverse transportation systems, for example, urban street networks [1–10], public transport networks [11–15], rail and subway networks [16–21], highway networks [22–24], and airline networks [25–29]. These studies are helpful for analyzing the structural characteristics of transport networks and the development of transportation science and engineering applications. Urban road network evolution considers the temporal dimension when studying the characteristics of an urban road network structure. Meanwhile, geographic information science provides many technical tools for the study because transport networks are a kind of spatial network.

Transportation is a minor subject in the system, and it is difficult to obtain data, especially in China; thus, research on the evolution of road networks is still limited. In particular, existing research mainly focuses on big and developed cities in the world such as Paris, London, Milan, Beijing, Shanghai, Hong Kong, and so on [30–36]. There are few reports on the analysis of medium-sized developing cities, such as Changchun studied in this paper. Meanwhile, existing studies proposed many measures to characterize road networks [37,38], and some of them found a link between the structure and function of road systems from the perspective of comparative studies of different cities and areas [39–43]. However, a temporal comparative analysis of a city in different periods has been seldom conducted to analyze the correlation between the structure and function of road networks. The study on the evolution of a city's road network might be beneficial for understanding the structural characteristics and evolution of the whole urban road network from a macroscopic perspective, building an urban road network evolution model to simulate its evolution process and mechanism, and providing a theoretical basis for solving related urban traffic problems.

Moreover, the city chosen for this study, Changchun, is a typical case among Chinese cities and has undergone a particular development period. Before 1949, city planning was largely based on what was learned from the advanced urban planning theories and methods of Japan and the former Soviet Union. The former Soviet Union built the modern rail system for Changchun. The Japanese formulated and implemented new urban planning during the Second World War (1931–1945). In the early years of new China (1949–), most Chinese cities were planned and rebuilt based on the old cities. Planners learned from the achievements of the ancients and respected the history and culture of the cities. As a young city that covers only over 200 years, Changchun has a relatively complete modern urban planning trajectory and road network data. Therefore, it is of high theoretical and practical value to analyze the evolution characteristics of the urban road networks in Changchun, China.

The paper is organized as follows. In Section 2, we will review existing studies. In Section 3, we will introduce the historical background of city construction and planning in detail, and then we will determine the sample years for subsequent research. In Section 4, we will propose the method of network modeling, analysis measures including state measures and function measures, and detailed analysis procedures. Section 5 will present the research results and analysis. Section 6 is our conclusions and discussions.

## 2. Literature Review

Existing studies in this field are limited, and most of them focus on metropolises or developed cities. For instance, Strano et al. reported an empirical analysis of almost 200 years of evolution of the road network in Milan (Italy) and found that the growth of the network was governed by two elementary processes: densification and exploration [30]. Masucci et al. explored the dynamic growth characteristics of greater London over the past two centuries (1786–2010) and pointed out that its growth could be predicted more accurately [31]. Barthelemy et al. systematically analyzed the evolution model of Paris street networks in the two periods of self-organization and planning (1789–2010) [32] and presented the data set of French roads and cities in the 18th century [44]. Masucci et al. studied the growth characteristics of the London street network using a dual method, proposed a continuous negotiation principle to extract the dual graph, and they found that logistic laws can describe the network growth, and the topological properties are mastered by the stable lognormal distribution [45,46]. Miao analyzed the evolution process and characteristics of Beijing's road networks

from 1969 to 2008, revealing the dense and expanding characteristics of the urban road network evolution [33]. Kang et al. studied the topological evolution characteristics of the road network in Shanghai Pudong new district (1995–2007), a region developed under the guidance of the government, using a road name based dual approach. They found that the urban road network has evolved from a broad-scale system to a scale-free system [36]. Lan used fractal theory and complex network theory to quantitatively describe the evolution characteristics and laws of the road network in Hong Kong from 1971 to 2011, from the perspectives of morphology and structure, and they constructed a road network evolution model. The findings were as follows: its geometry has always presented a dual fractal nature and a core-edge pattern during the development of the Hong Kong road network in the past 40 years. In terms of structure complexity, the average shortest path length and clustering coefficient increased year by year [35]. Based on previous research results [31,45,46], Masucci et al. analyzed the robustness and centrality of self-organized and planned cities in London and Chicago and found that the planned city of Chicago was superior to London to some extent [47]. Casali et al. analyzed the topological growth in the Zurich (Switzerland) road network and found changes over time [34].

Overall, these existing studies have several limitations: limited numbers, their focus on only metropolises or developed cities, and a lack of evidence on the correlation between structure and function of road networks from the perspective of evolution. Thus, it is vital to conduct research on the analysis of road network evolution based on Changchun, China, which has almost complete road data since its founding. The contributions of this study mainly follow three aspects.

(1) This work analyzes the evolution pattern of a middle-sized developing city, which has been reported on rarely. Besides the evolution pattern, the study framework and methodological analysis workflow based on network theory proposed here could be applied to the evolutionary study of road networks in almost any city.

(2) This work proposes a characterization of network scale evolution and the network measure evolution from the perspective of multidisciplinary integration, including network science, geographic information science, and transport science. The characteristics of road network evolution could help us define the period of road development. On this basis, the network measure results could help optimize road network performance, such as improving the network's efficiency and reducing the average travel distance.

(3) This work determines whether transparent relationships between structure and function of the road networks exist. The study will provide empirical evidence on the correlation from the perspective of evolution. After we validate the correlation, this network theory will be more helpful for transport studies in the future.

## 3. Data Collection

Changchun is one of the major cities in northeast China, a famous national historical and cultural city, and a comprehensive transportation hub in China. It is located in north latitude 43°05′~45°15′, longitude 124°18′~127°05′ and was selected as the case city of the study. When we study the evolution characteristics of urban road networks, it is necessary to acquire and analyze primary urban road network data from different ages. Therefore, how to select and determine the time series of the research object is the first problem to be solved. The form and evolution trend of the urban road networks are firmly related to specific historical periods and planning concepts, so we will first analyze the different development and planning periods of Changchun.

### 3.1. Historical Periods and Road Network Digitization

This section gives a brief introduction on the different historical periods of Changchun and uses ArcGIS software (ESRI, RedLands, USA) to carry out manual digitization of the maps to intuitively display the road traffic networks in different historical periods.

Changchun is a young city developed in modern times with a history of only over 200 years. In 1800, Changchun Hall was formally established by the Qing Dynasty government (1616–1912). Until the end of the 19th century, Changchun was just an ordinary frontier market town, still in the urban formation period [48]. So, the urban road networks of the 19th century were not analyzed here. In the 20th century, between 1904 and 1905, the Japanese empire had an imperialist war with tsarist Russia on the land of northeast China to fight for the control of China's Liaodong peninsula and the Korean peninsula, which failed in tsarist Russia. After the war, a treaty on matters concerning the three northeastern provinces was signed, which required the Qing government to develop Changchun, Harbin, and other14 cities in northeast China as trading ports of Japan to sell goods [49]. An example of the road network in this period is shown in Figure 1. To facilitate the construction of the road network topology in Gephi, the nodes in the figure represent intersections and dead ends. Meanwhile, to save space, the road networks of other periods are not shown in this section, and they will be shown in a complete map in Section 5.

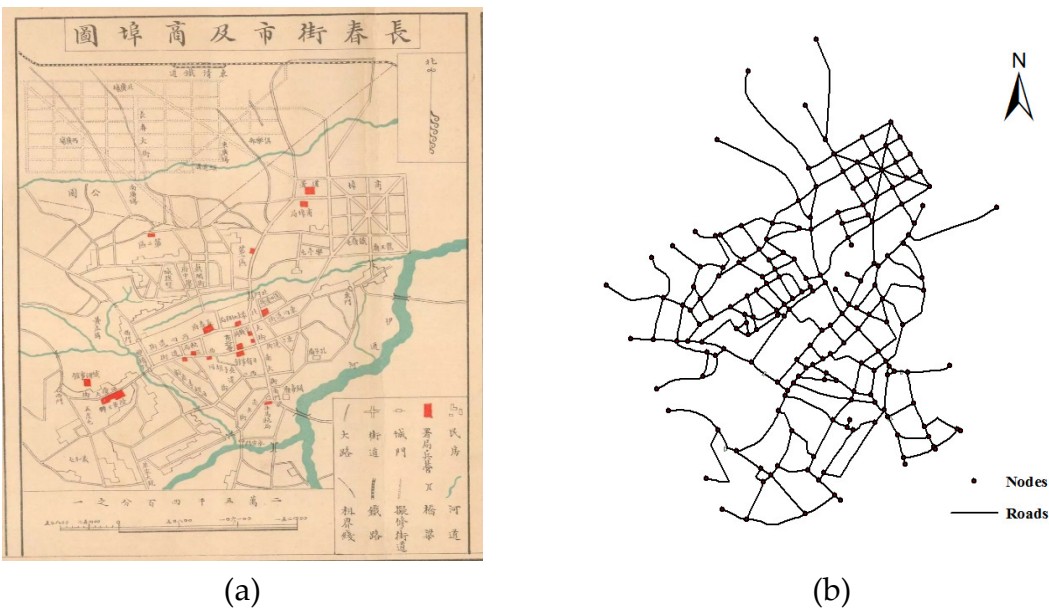

(a)　　　　　　　　　　　　　　　　　　　　　　　　　(b)

**Figure 1.** The road network of Changchun in 1912. (**a**) Original traffic map; (**b**) vector road network.

After steady development, Changchun became a colony of the Japanese imperialists in 1931. A puppet regime, Manchukuo, was established in 1932 under the manipulation of Japanese imperialism to rule China, and Changchun was designated as the capital and renamed Hsinking [50]. During the period of Manchukuo (1932–1945), the Japanese colonists formulated and implemented new city planning, which influenced the future development of Changchun's road networks. In this planning, the single-center layout was adopted to define Renmin street as the central axis of the city, which further strengthened the urban network structure of "axis + garden square + radiation + grid", and ring roads were set up in the periphery of the city [48]. After the first phase (March 1932 to December 1937) of new planning and construction, the road network pattern was different from the old city and began to shape the distribution pattern of the current road network. However, in the later stage of construction, the Second World War broke out. Because of the tense war situation and the shortage of funds, the construction of Hsinking gradually came to a halt, and part of the plan was not realized. During the war of resistance against Japan (1931–1945) and the war of liberation (1945–1949), urban construction of Changchun was basically at a standstill, and 60% of municipal facilities were damaged [49]. With the foundation of the People's Republic of China, Changchun city entered a wave of reconstruction and a period of recovery. After years of restoration and reconstruction, the city had regained its vitality, and the social infrastructure had been improved day by day. In the 1950s, Changchun set up an urban planning group and carried out comprehensive urban planning (1952, 1953, 1957) [48].

Between 1958 and 1978, the Great Leap Forward (1958–1960) and the Cultural Revolution (1966–1976) broke out in China, and the two incidents affected the urban planning significantly. In 1960, Changchun implemented policies of the national planning conference to not carry out planning, downsizing, and merging of planning agencies. In 1966, Changchun abolished the planning agency, and urban planning of the city stalled [48]. After the end of the civil unrest, the government of Changchun began to work out the urban master planning of the provincial capital in 1980 to deal with a series of problems encountered in urban construction in the new era. In 1983, the government of Jilin province in China submitted to the state council the report on the application for approval of the master urban plan of Changchun. In 1985, the state council of China approved and pointed out that all new construction and reconstruction should be carried out in strict accordance with the plan. Subsequently, the master plan of 1980 was fine-tuned to expand the size of land and population. From 1985 to 1993, the urban construction of Changchun developed rapidly. The Changchun economic development zone was formally established in 1992, and the high-tech development zone was approved in 1993. The road network of Changchun city at that time could no longer meet the needs of urban development, so the Changchun city master plan (1996–2010) was prepared, which determined development of the new district as the main task and reconstruction of the old district as the supplement task [48]. In 2003, Changchun completed the leading construction and development goals of the 1996 urban planning ahead of schedule. At the same time, a new round of revisions of the master plan was officially launched in 2004 to meet the strategic needs of building a well-off society in an all-around way and revitalizing the old industrial base in northeast China. The plan was submitted to the State Council in 2006. On 26 December 2011, the state council approved the urban master plan of Changchun (2011–2020), which provided practical guidance and a legal guarantee for the rapid development of Changchun in the new era. To further implement the new round of the northeast revitalization strategy and improve the quality and level of the new urbanization, the Changchun government revised the original plan in 2017 and formed the urban master plan of Changchun (revised in 2017). In the revision, it is proposed to perfect the existing road system and form an efficient and convenient urban road network. The city is planned to form one ring with three horizontal and four vertical urban expressway systems in the central urban area. With this as the skeleton, the primary and secondary trunk roads and branch roads combine to form an urban road network system with a grid network as the main form [48].

Note that the road network scopes of the maps for different years are not the same. To reflect more comprehensively the road evolution of Changchun, we determined that the research focus should be the road network within the area of Changchun ring expressway, and part of regular roads outside the ring expressway are appropriately added, but the highway was not included.

*3.2. Determining the Sampling Years*

As a young city developed in modern times with a history of only over 200 years, the urban planning history of Changchun city can be roughly divided into six stages according to the development history mentioned above in Section 3.1. The evolution of urban road networks is closely related to urban planning. We collected the traffic maps with the help of the Changchun Institute of Urban Planning and Design, and the earliest map was 1912. Meanwhile, we noticed that the network size of Changchun road networks in 1912 was quite small. In 1912, there were only 187 nodes (including 161 intersections and 26 dead ends) and 284 edges. Therefore, it is acceptable to study the network evolution from 1912 to 2017. The starting and ending years of different historical stages are almost covered according to the sampling frames including the early maps (1912–1992) provided by Changchun Institute of Urban Planning and Design and the collected maps in recent years (1998–2017). Table 1 shows the road network evolution stages and research years that roughly cover the complete life cycle of urban development in Changchun.

**Table 1.** Six stages and research years of the urban road network evolution in Changchun city.

| Periods | Time Range | Sampling Years | Planners | Age characteristics |
|---------|-----------|----------------|----------|---------------------|
| Stage 1 | 1800–1897 | Not considered | Qing Dynasty | The early stage of urban construction |
| Stage 2 | 1898–1931 | 1912, 1932 | | Early stage of modernization |
| Stage 3 | 1932–1945 | 1932, 1939, 1946 | Japanese | Rapid development period of modernization (Hsinking) |
| Stage 4 | 1946–1957 | 1946, 1958 | | Completion period of modernization |
| Stage 5 | 1958–1977 | 1958, 1968 | Chinese | Restoration and reconstruction period |
| Stage 6 | 1978 to now | 1983, 1992, 1998, 2006, 2011, 2017 | | Rapid urban construction period |

## 4. Methodology

### 4.1. Network Modeling

A road network based on the graph or network theory consists of nodes and edges. The primal approach and dual approach are two kinds of network modeling methods that are widely applied in existing studies [4,5]. When authors use open geographic data, such as OpenStreetMap, intersection counts are overestimated due to artificial breakpoints, and the average edge lengths are underestimated in many studies [51]. In this paper, we will study measures of urban street patterns of different periods in Changchun represented as a spatial network. To avoid the problem identified by Boeing, the study will use a primal approach to model the urban road center-line network. Namely, the intersections and dead ends, not false nodes, are regarded strictly as the nodes, and the road segments are the edges [4,8]. Figure 2 is a schematic diagram of the primal approach. The approach works within a fully metric framework in which distance has to be measured in a proper spatial term (meters or miles), not just in topological terms (steps).

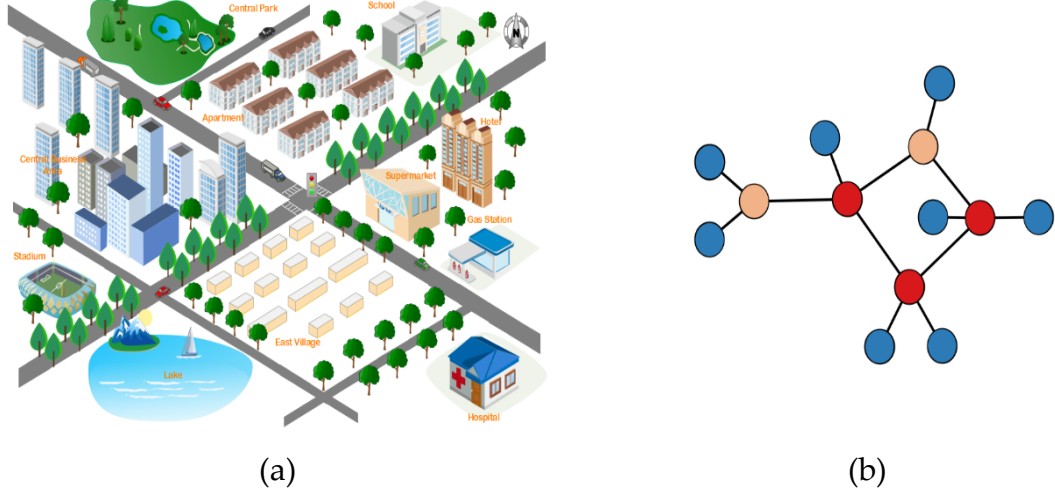

(a)                                                                      (b)

**Figure 2.** Schematic diagrams of an urban road network. (**a**) Real road network; (**b**) primal approach modeled network, degree values increase from blue to red.

### 4.2. Research Measures for Analysis

#### 4.2.1. State Measures

State refers to the current development phase of a road network structure, whether it is a relatively simple network or a more complex one [20]. Three indicators developed by Garrison and Marble are first introduced here: $\alpha$ index, $\beta$ index and $\gamma$ index [52,53].

If $v$ is the number of vertices, $e$ is the number of edges. A graph is a collection of $v$ vertices and $e$ edges. In urban road networks, the vertices are the intersections and the dead ends, and the edges are the road segments. $v$, $e$, and road total length show the size or scale of the road networks.

$$\alpha_{planar} = \frac{e - v + 1}{2v - 5}. \tag{1}$$

$\alpha_{planar}$ is the $\alpha$ index for planar graphs. The urban road networks here were regarded as planar networks.

$$\beta = \frac{e}{v}. \tag{2}$$

The $\beta$ index or complexity is the ratio of the number of edges $e$ and the number of vertices $v$, which can be interpreted as the average number of edges per vertex. It is bound between 0 and $+\infty$.

$$\gamma_{planar} = \frac{e}{3v - 6}. \tag{3}$$

$\gamma_{planar}$ is the $\gamma$ index for planar graphs. The $\gamma$ index or degree of connectivity is the ratio of the actual number of edges $e$ and the potential number of edges in a completely connected graph. It is bound between 0 and 1 but can be expressed as a percentage [54].

The fractal theory is an excellent tool to describe complicated geographic objects [55], such as the kind of urban road network in this study [56]. The definition of fractal dimension, which is a character variable, is as follows [57].

$$N = \left(\frac{S}{s}\right)^{D}, \tag{4}$$

where $D$ is the fractal dimension. In the equation, $s$ is the size of squares to cover an urban road network in the box-counting method. The grid that covers the entire road network is assigned size $S$. Then, the number of squares in which a part of the road network is counted is $N$.

Complex network studies provide significant measures to analyze the urban road network. Degree and clustering coefficients are traditional measures in complex networks. As a fundamental parameter to describe the characteristics of a complex network, the degree $k_i$ of a node $i$ is defined as the number of nodes connected to it [58]. The higher the degree of a node, the more critical it is. Average degree $\langle k \rangle$ is the average value of all node degrees.

$$\langle k \rangle = \frac{\sum\limits_{i=1}^{v} k_i}{v}. \tag{5}$$

The clustering coefficient is the ratio of the number of connected edges to the maximum number of edges. Assuming that the node $i$ is directly connected to $k_i$ nodes in the undirected network, the maximum number of possible edges between the $k_i$ nodes is $k_i(k_i - 1)/2$, while the actual number of edges is $M_i$. Then, the clustering coefficient $C_i$ of node $i$ is

$$C_i = \frac{2M_i}{k_i(k_i - 1)}. \tag{6}$$

The average clustering coefficient refers to the average value of clustering coefficients of all nodes in the graph.

$$\langle C \rangle = \frac{1}{N} \sum\limits_{i=1}^{v} C_i, \tag{7}$$

where $0 \leq \langle C \rangle \leq 1$. When $\langle C \rangle = 0$, all nodes are isolated nodes without connected edges. When $\langle C \rangle = 1$, the network is a complete graph of all nodes with connected edges between any two nodes.

### 4.2.2. Function Measures

Many measures represent the physical function of the road network, such as path length, efficiency, choice, and integration. In the network, the number of edges of a path that has the least number of edges between two nodes $i$ and $j$ is called the shortest path length $d_{ij}$. The average path length $L$ is defined as the average distance between all node pairs.

$$L = \frac{1}{v^2} \sum_{j=1}^{v} \sum_{i=1}^{v} d_{ij}. \tag{8}$$

In the undirected network, $d_{ij} = d_{ji}$, $d_{ii} = 0$. Equation (4) is simplified as

$$L = \frac{2}{v(v-1)} \sum_{i=1}^{v} \sum_{j=i+1}^{v} d_{ij}. \tag{9}$$

If we consider road length as the weight of each edge, we can get the weighted average path length.

$$E = \frac{1}{v(v-1)} \sum_{i \neq j} \frac{1}{d_{ij}}. \tag{10}$$

$E$ is the efficiency of a network [59], where $d_{ij}$ is the shortest path length between the vertex $i$ and $j$. We will discuss distance-based efficiency in the paper.

Space syntax explores the logical relationship between space, describes the space configuration quantitatively, and finds the relationship between spatial form and human society [60]. In theory, choice and integration are two fundamental indicators.

The choice is a measure to calculate the probability or number of the shortest path of an axis or street from one space to all other spaces, which is the number spaces that appears in the shortest topology path. If $e_{ij}$ is an edge between the node $i$ and $j$, then the choice is described as follows:

$$B_{ij} = \sum_{l \neq m} [n_{lm}(e_{ij}) / n_{lm}], \tag{11}$$

where $n_{lm}$ represents the number of shortest paths between the node $l$ and $m$, and $n_{lm}(e_{ij})$ represents the number of shortest paths that pass the edge $e_{ij}$ between the node $l$ and $m$. The choice is the same as the betweenness of complex networks.

The average depth $MD_i$ represents the average distance between node $i$ and all other nodes.

$$MD_i = \frac{1}{v-1} \sum_{j=1}^{v} d_{ij}, \tag{12}$$

where $d_{ij}$ is the shortest distance between the node $i$ and $j$, which is the total depth within the space region. It is consistent with the meaning of the shortest path length in a complex network.

Integration $I_i$ refers to the standardized distance from any space to all other spaces in the system. The distance from the starting space to all other spaces can be calculated with the following equation [61]:

$$I_i = \frac{(MD_i - 1)(v - 1)}{v[\log_2((v+2)/3 - 1) + 1]}. \tag{13}$$

### 4.3. Analytical Procedures

The study framework of road network evolution and analysis is shown in Figure 3. After we collected traffic maps in different years, the vector road networks were generated by ArcGIS digitization,

and then they were projected to generate the length and area of road segments so that we could analyze the changes of road mileages in different years. The spatial join toolbox of ArcGIS was used to generate the connection relation of vertices and edges. Then, the data table of edges, including the connection and weight, as shown in Table 2, was created and imported into Gephi. Figure 4 shows the analysis flow based on the study framework in the way of road network data visualization. The whole study was based on network measures, which are shown on the right side of each section. Weighted path length and efficiency were derived from the OD (origin and destination) cost of the network analysis in ArcGIS. Gephi software could generate the network graph and calculate the necessary network measures for comparative analyses. Vector road networks could be converted into Depthmap. Then, weighted choice and integration could be calculated. The fractal dimension was attained by the box-counting method with the help of ArcView. ArcGIS (https://www.esri.com) is a mapping and analytics platform widely used in geographic information science. It provides contextual tools for mapping and spatial reasoning. ArcView is a part of ArcGIS. Depthmap is an open source and multiplatform spatial analysis software for spatial networks of different scales. The software was originally developed by Alasdair Turner at UCL(University College London). Gephi (https://gephi.org/) is the leading visualization and exploration software for all kinds of graphs and networks. More importantly, it is open source and free.

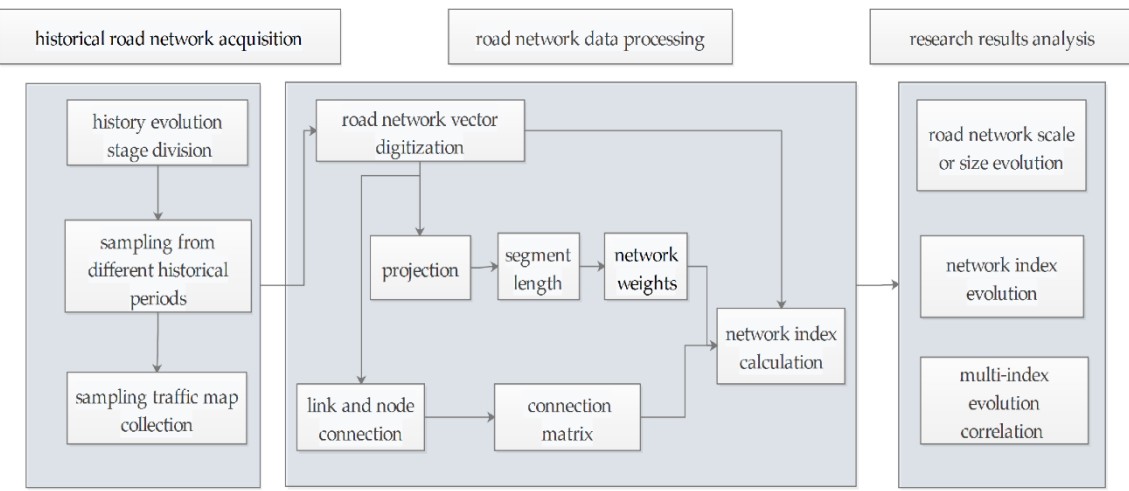

**Figure 3.** The study framework of road network evolution and analysis.

**Table 2.** An example of edge data imported into Gephi.

| Source | Target | Label | Weight | Type |
|--------|--------|-------|----------|------------|
| 72 | 67 | 0 | 260.1685 | Undirected |
| 167 | 72 | 1 | 1138.937 | Undirected |
| 67 | 68 | 2 | 144.8865 | Undirected |
| 68 | 80 | 3 | 608.7503 | Undirected |
| 80 | 81 | 4 | 104.1086 | Undirected |

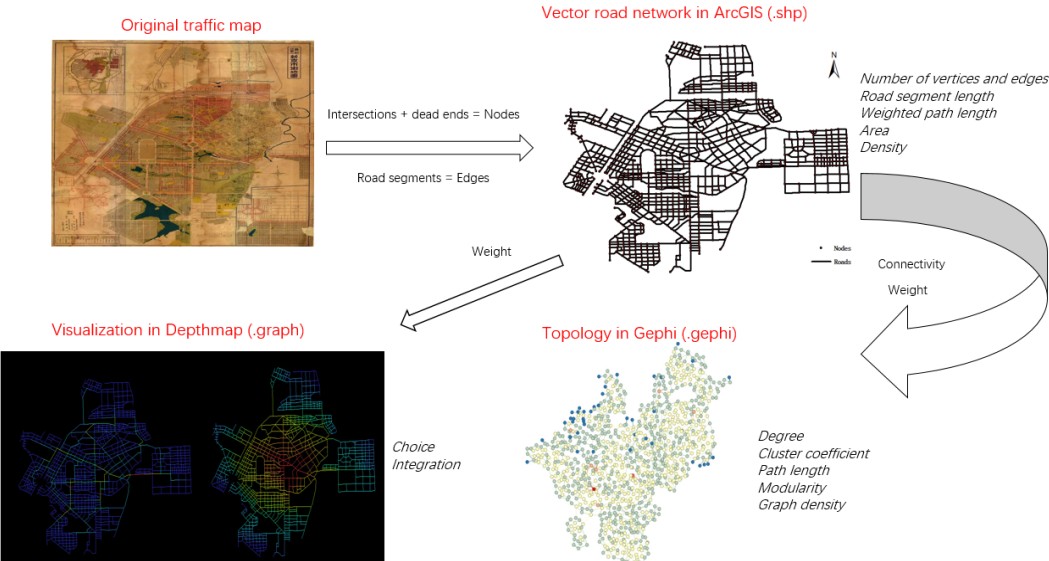

**Figure 4.** Visualization of the methodological workflow in the study based on the road network in 1939.

## 5. Analysis Results

To scientifically and intuitively present the changes of road networks in Changchun from the initial stage of construction to the current period, this section will quantitatively analyze its growth and evolution characteristics from the perspectives of different network indicators and correlation.

The road networks of 12 different historical periods in Changchun were summarized, and they basically cover the complete development cycle of Changchun. Figure 5 shows the evolution of road networks in Changchun from 1912 to 2017, which directly shows the changes in spatial location and scale of the urban road networks in different time series. Changchun's urban road networks had two visible evolution characteristics: on the one hand, the road network covered a growing area and expanded outward continuously; on the other hand, there were more and more road segments within the ring road. This is consistent with the results of studies of other cities [34,62]. However, considering the development stages, it is different from other cities that the Changchun road network presents a cyclical progressive evolution. Taking the Great Leap Forward (1958–1960) and the Cultural Revolution periods (1966–1976) as the boundary, development of the Changchun road network could be divided into two different periods: the Japanese planning period (1932–1958) and the Chinese planning period (1968–2017), but the latter relies on the primary road network planned by the Japanese in the Second World War. From 1945 to 1958, many roads were damaged and unsurfaced. Built asphalt roads were shown in the 1968 road network, so the network scale was different from the road network of 1958.

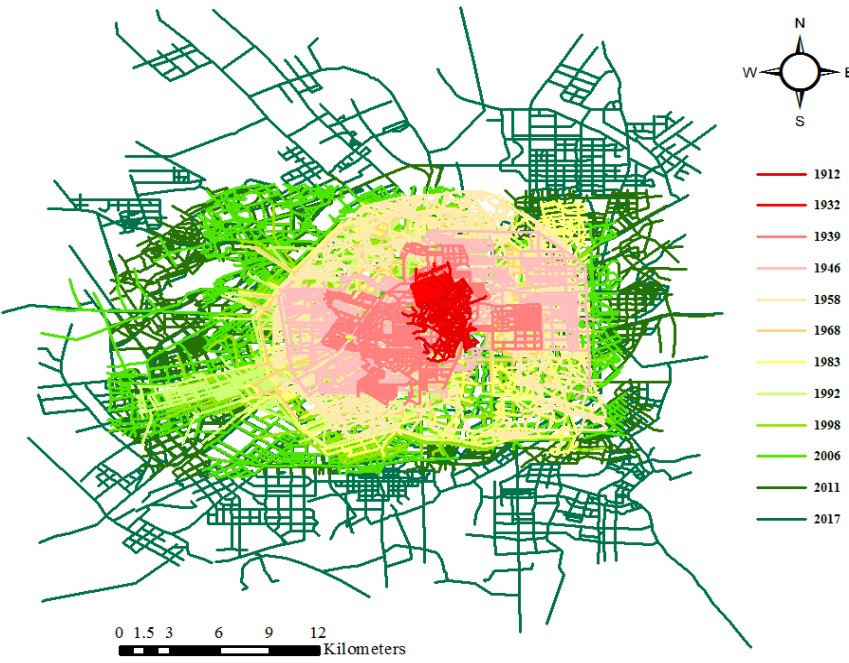

**Figure 5.** Road network evolution in Changchun, China (1912–2017).

## 5.1. Road Network Scale

By comparing the road network vector data of 12 different years in Changchun, we can intuitively see that the urban road network continuously expanded and regularly compacted. When the network scale was analyzed quantitatively, the evolution of road networks in Changchun city was notably different from that of other cities, which has significant historical characteristics. The rapid development of urban roads includes the construction of Hsinking in Manchukuo and the reform in, and opening up of, the People's Republic of China. Between the two periods, the urban roads were severely damaged, affected by the Second World War and the liberation war of China. After postwar recovery and reconstruction, road conditions gradually improved, and network construction developed rapidly under good social conditions and national policies. Figure 6 shows that the number of nodes and edges and the length of roads exhibited a trend of steady growth from the initial stage of the city's construction to Manchukuo, a sharp decline due to postwar replanning, as well as steady growth after the reform and opening up of new China. The difference in the three indicators is that the number of intersections and segments gradually stabilized and decreased after 2006, but the road mileage still increased.

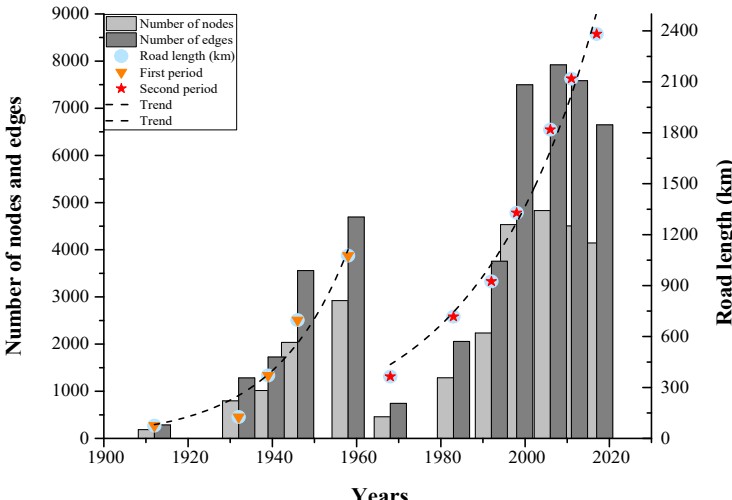

**Figure 6.** Changes in the number of nodes and edges, and road total lengths from 1912 to 2017.

When investigating the relationship between the number of intersections, the number of road segments and the road length, we found that there was a significant, linear correlation among them. The correlation coefficient between the number of intersections and road segments was as high as 0.9999. There was also a significant correlation between road mileage and the number of intersections or roads, with correlation coefficients of 0.9160 and 0.9172, respectively. OriginPro provided favorable results of significance tests for our study. The correlation, as shown in Figure 7, shows that the macroscale development of Changchun's road system was relatively stable. The relationship between the number of nodes and road mileage was the same as the results of American cities [62]. The ratio of the number of segments to the number of intersections, which was relatively stable, ranged from 1.73 to 1.82 with an average value of 1.77. The $\beta$ index, including the dead ends, was slightly less than the ratio mentioned above. It was bound between 1.52 and 1.74, and the mean was 1.64. The basic information of road networks in different years is shown in Table 3. $\alpha$, $\beta$, and $\gamma$ showed significant, linear correlations among them, and the correlation coefficients were higher than 0.99. The urban road networks planned by the Japanese (1939, 1946) had higher $\alpha$, $\beta$, and $\gamma$. These three indicators had limited benefits to characterize the road networks, which was the same as the findings in Zurich [34].

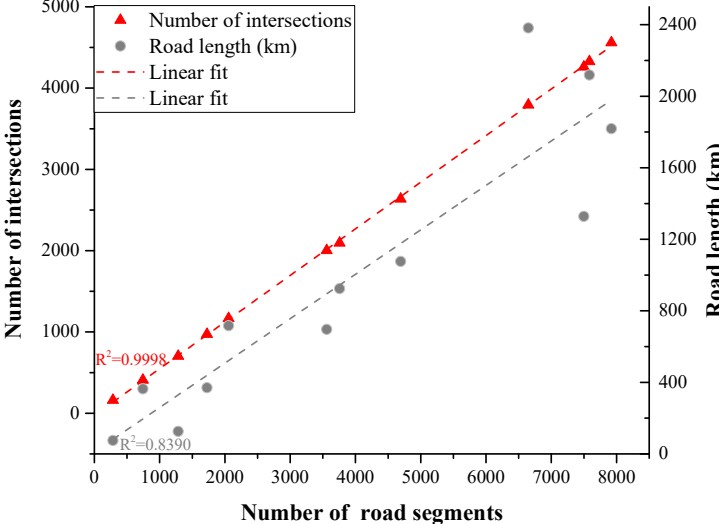

**Figure 7.** Correlation between the number of road segments, intersections, and road lengths.

**Table 3.** Basic information on road networks for different years in Changchun.

| Years | Nodes | Edges | $\alpha$ | $\beta$ | $\gamma$ |
|---|---|---|---|---|---|
| 1912 | 187 | 284 | 0.2656 | 1.5187 | 0.5117 |
| 1932 | 797 | 1284 | 0.3071 | 1.6110 | 0.5384 |
| 1939 | 1015 | 1726 | 0.3516 | 1.7005 | 0.5679 |
| 1946 | 2037 | 3559 | 0.3743 | 1.7472 | 0.5830 |
| 1958 | 2923 | 4693 | 0.3032 | 1.6055 | 0.5355 |
| 1968 | 459 | 744 | 0.3133 | 1.6209 | 0.5427 |
| 1983 | 1285 | 2057 | 0.3014 | 1.6008 | 0.5344 |
| 1992 | 2236 | 3757 | 0.3407 | 1.6802 | 0.5606 |
| 1998 | 4534 | 7499 | 0.3273 | 1.6539 | 0.5516 |
| 2006 | 4830 | 7920 | 0.3201 | 1.6398 | 0.5468 |
| 2011 | 4505 | 7584 | 0.3420 | 1.6835 | 0.5614 |
| 2017 | 4143 | 6648 | 0.3026 | 1.6046 | 0.5351 |
| Mean | 2412.58 | 3979.58 | 0.3208 | 1.6389 | 0.5474 |
| Std. Dev. | 1725.59 | 2843.31 | 0.0286 | 0.0593 | 0.0189 |

The statistical bulletin of national economic and social development of Changchun Bureau of Statistics gives the statistical data of urban road length, area, population (from 2001 to now), and private car ownership (from 2009 to now) from 2000 to now. The bulletin has recorded the urban development of this city in detail since 2000 [63], and the urban road network in this bulletin includes not only the main urban area but also the surrounding areas and counties. In terms of the development and evolution of urban roads, the overall region of the city has been in a stage of steadily rising development since the beginning of the 21st century. The length and area of roads have gradually increased with time, and the linear relationship is apparent, as shown in Figure 8a. The population merely covers seven municipal districts and does not include Nongan county nor the cities of Yushu and Dehui. From 2014 to 2015, the population increased dramatically, as the light blue line in Figure 8a shows. Meanwhile, data analysis results show that there was a significant, linear correlation between urban road development (length and area) and time, indicating a strictly planned city development style. The road length and private car ownership exhibited a linear correlation with the urban population, and the correlation coefficients were greater than 0.85, as shown in Figure 8b. The internal logic is that travel demand and the demands for roads and private cars increased with the increase of the urban population. This is a symbol of development and growth for a city.

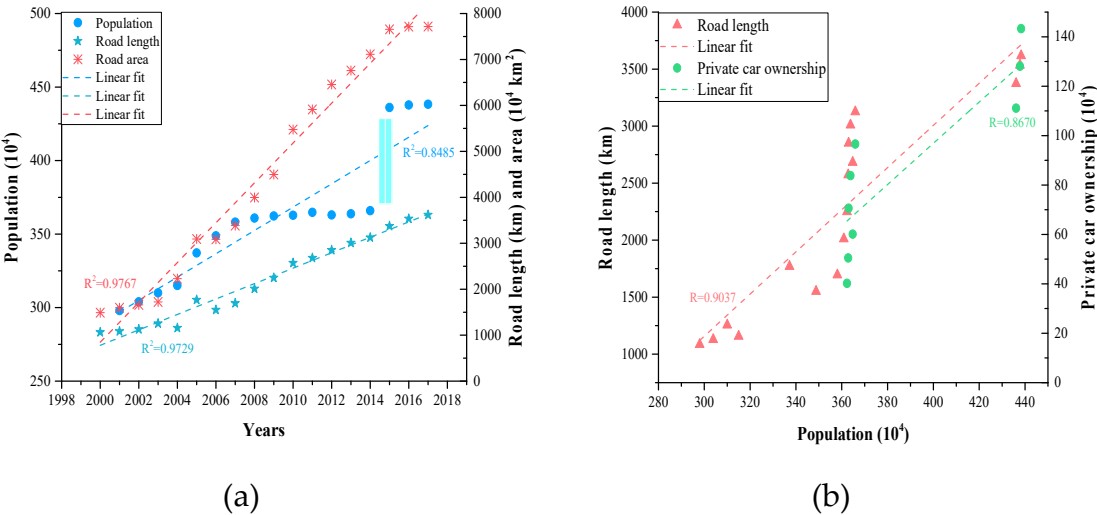

**Figure 8.** Urban expansion with time. (**a**) The evolution of the population, road length, and area; (**b**) the population's correlation with road length and car ownership.

### 5.2. Network Measure Evolution

The data of the edge were imported into Gephi to generate a network diagram. In Gephi, a statistical function was run to calculate various complex network indicators, and the layout was set as ForceAtlas 2 after 4 h to stabilize the overall layout. The following road network topology and structural indicators from 1912 to 2017 can be obtained. Depthmap is used to visualize the probability and global distance of each segment in the shortest paths. Here, we took the road network of 1912 as an example to show the topology (Figure 9) and parameters of the space syntax (Figure 10).

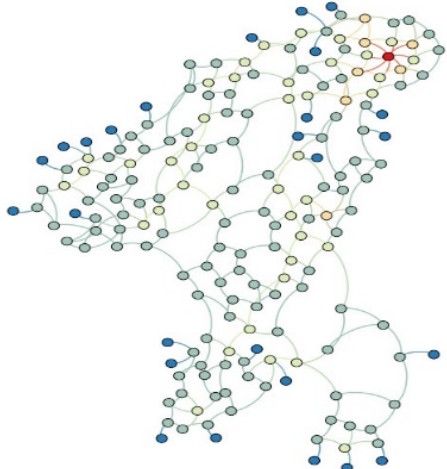

**Figure 9.** The topology of Changchun in 1912. Degree values increase from blue to red.

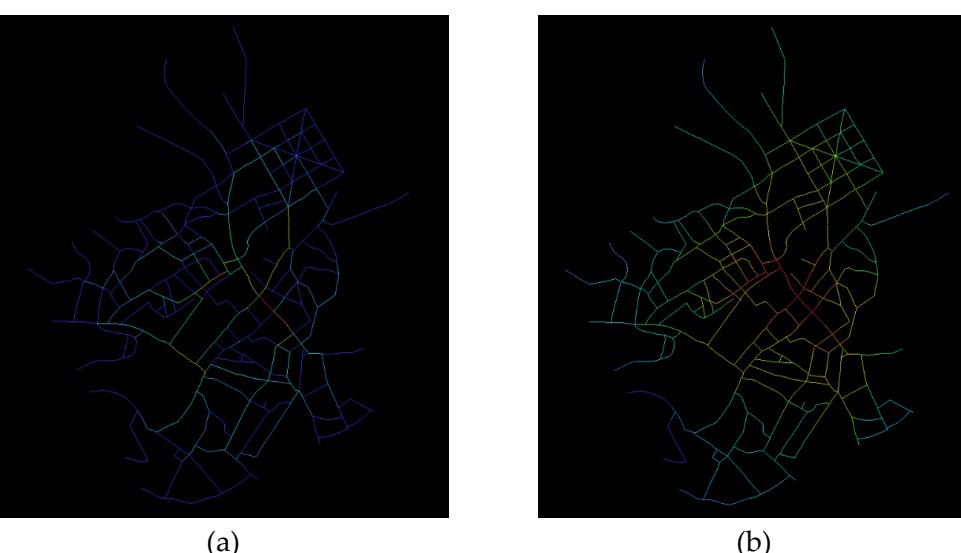

(a)                                         (b)

**Figure 10.** Space syntax parameters in 1912. (**a**) Choice; (**b**) integration. Values increase from blue to red. The figure shows the difference between the roads in the same period. Other years are available upon request.

The degree, path length, and cluster coefficient are three fundamental indicators in network science, so they were analyzed first here. The frequency distribution and evolution of the node degree with time are shown in Figure 11. The frequency of different degree values represents the number of different types of intersections. The node with a degree 1 represented a cul-de-sac in the road network. In different years, the average proportion of the sum of the number of T-junctions and crossroads to the total number of intersections was 97.77% (excluding the dead ends). When the dead ends were included, the proportion was 90.36%. According to statistics, there is a specific relationship between the

number of dead ends and the number of different types of intersections, and the maximum correlation coefficient was 0.85 for the T-junctions. Meanwhile, besides the very rare intersections with node degrees of 7 and 8, the correlation between the number of intersections with different types was also apparent. The average correlation coefficient among the number of these four types of intersections (node degrees of 3, 4, 5, and 6) in any year was 0.9718, indicating a steady development trend. Except for 1946, during the Japanese occupation, the T-junction accounted for the most substantial proportion in each period, which was still consistent with the situation in American cities [57]. The frequency of the degree of 3 was very close to that of the degree of 4 in 1939 and 1946, which shows a preference for the grid network in Japanese planning.

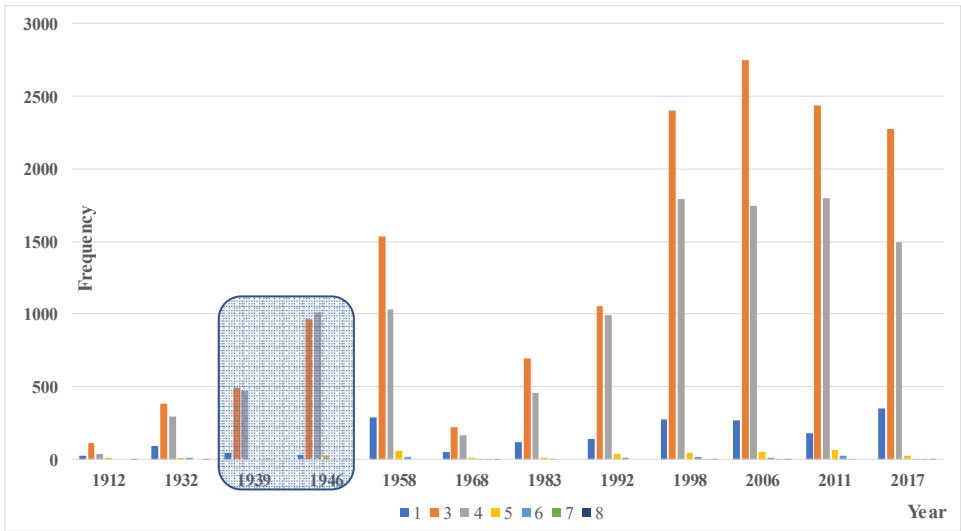

**Figure 11.** Node distribution of Changchun road networks in different years.

The average path length and clustering coefficient are two main indexes to determine whether it is a small world network [64,65]. The average path length represents the depth of the urban road transportation systems, which requires two nodes in the network to be accessible through as few edges as possible. For example, the average path length of the 1912 road network was about 8.69, which required at least nine road segments to connect any two intersections in the network. In the two planning cycles, the values of path length increased over time and then exhibited a steady state (i.e., it will not change a lot in the short term). The clustering coefficient represents the average clustering degree of the network formed by the intersections and the road segments and is a symbol of the width of the network. For example, the clustering coefficient of the 1912 road traffic network was about 0.089, which had a significant clustering coefficient compared to other years. The trend of the clustering coefficient seems to be opposite to that of path length. If the network has a smaller average path length and more significant clustering coefficient, namely conforming to the following two equations, it is a small-world network.

$$L \geq L_{random}, \tag{14}$$

$$\langle C \rangle \gg \langle C_{random} \rangle, \tag{15}$$

where $L_{random}$ and $C_{random}$ represent the average path length and clustering coefficient of a random network with the same scale, which can be described as follows [65]:

$$L_{random} = \ln N / \ln \langle k \rangle, \tag{16}$$

$$C_{random} = \langle k \rangle / N. \tag{17}$$

In network science, a small-world network is regarded to have a better structure for travel than a network without small worldliness. Table 4 shows the values of the two indicators and the examination results of small worldliness for the Changchun road networks. Therefore, the road networks of Changchun in different selected years were all small-world networks.

**Table 4.** Examination of small worldliness.

| Years | $L$ | $C$ | $L_{random}$ | $C_{random}$ | $L \geq L_{random}$? | $C >> C_{random}$? | Small World? |
|---|---|---|---|---|---|---|---|
| 1912 | 8.69 | 0.089 | 4.72 | $1.62 \times 10^{-2}$ | Yes | Yes | Yes |
| 1932 | 18.25 | 0.057 | 5.71 | $4.04 \times 10^{-3}$ | Yes | Yes | Yes |
| 1939 | 20.05 | 0.043 | 5.66 | $3.35 \times 10^{-3}$ | Yes | Yes | Yes |
| 1946 | 25.80 | 0.030 | 6.09 | $1.72 \times 10^{-3}$ | Yes | Yes | Yes |
| 1958 | 29.02 | 0.067 | 6.84 | $1.10 \times 10^{-3}$ | Yes | Yes | Yes |
| 1968 | 12.54 | 0.067 | 5.21 | $7.06 \times 10^{-3}$ | Yes | Yes | Yes |
| 1983 | 19.45 | 0.057 | 6.15 | $2.49 \times 10^{-3}$ | Yes | Yes | Yes |
| 1992 | 23.36 | 0.041 | 6.36 | $1.50 \times 10^{-3}$ | Yes | Yes | Yes |
| 1998 | 34.43 | 0.036 | 7.04 | $7.30 \times 10^{-4}$ | Yes | Yes | Yes |
| 2006 | 31.85 | 0.036 | 7.14 | $6.79 \times 10^{-4}$ | Yes | Yes | Yes |
| 2011 | 32.39 | 0.040 | 6.93 | $7.47 \times 10^{-4}$ | Yes | Yes | Yes |
| 2017 | 28.86 | 0.031 | 7.14 | $7.75 \times 10^{-4}$ | Yes | Yes | Yes |

Other structural measures are shown in Table 5. These three indicators—fractal dimension, average degree, and modularity—did not change much with time. The maximum variation range was 17.17 of the fractal dimension from 1992 to 1998, and it was less than 5% in most cases. Another group of indicators including efficiency, density, segment length, and weighted path length had an extensive range of changes, usually more than 20%.

**Table 5.** Structural indicators of Changchun road networks in different years.

| Years | Fractal Dimension | Degree | Modularity | Efficiency | Density (km/km$^2$) | Segment Length (m) | Weighted Path Length (km) |
|---|---|---|---|---|---|---|---|
| 1912 | 1.182 | 3.02 | 0.809 | $6.46 \times 10^{-4}$ | 5.08 | 529.12 | 2.38 |
| 1932 | 1.285 | 3.22 | 0.875 | $9.05 \times 10^{-4}$ | 11.35 | 195.84 | 1.99 |
| 1939 | 1.279 | 3.40 | 0.886 | $3.20 \times 10^{-4}$ | 5.87 | 428.82 | 5.09 |
| 1946 | 1.334 | 3.49 | 0.900 | $2.59 \times 10^{-4}$ | 5.71 | 391.21 | 6.44 |
| 1958 | 1.189 | 3.21 | 0.930 | $1.87 \times 10^{-4}$ | 5.26 | 458.43 | 8.82 |
| 1968 | 1.217 | 3.24 | 0.847 | $2.74 \times 10^{-4}$ | 2.36 | 977.21 | 6.21 |
| 1983 | 1.249 | 3.20 | 0.906 | $1.99 \times 10^{-4}$ | 2.95 | 696.43 | 8.66 |
| 1992 | 1.188 | 3.36 | 0.910 | $2.20 \times 10^{-4}$ | 3.86 | 246.07 | 7.23 |
| 1998 | 1.392 | 3.31 | 0.929 | $2.23 \times 10^{-4}$ | 4.66 | 354.21 | 7.02 |
| 2006 | 1.388 | 3.28 | 0.929 | $1.68 \times 10^{-4}$ | 4.34 | 459.15 | 9.38 |
| 2011 | 1.396 | 3.37 | 0.928 | $1.47 \times 10^{-4}$ | 4.35 | 279.35 | 10.58 |
| 2017 | 1.306 | 3.21 | 0.928 | $1.40 \times 10^{-4}$ | 4.30 | 358.26 | 12.37 |

Higher choice values of space syntax show the arterial roads that go through the shortest paths in different road networks. Integration of different road networks shows a trend of gradual expansion from the middle to the surrounding areas, which can clearly show the hierarchy of the local roads in the road networks. The non-blue roads in Figure 10a show the arterial roads in the road network, and these roads had shorter paths to pass. Although there are differences between the shortest path and the actual travel path of residents [66–68], the shortest path is still a commonly used path selection algorithm at present.

Since it is difficult to match the road names of the current road network for the early roads, the arterial roads in the road networks from 1939 to now were selected for analysis by a word cloud. The word cloud can display different sizes and colors according to the frequencies of different road names (i.e., keywords). The higher the frequency is, the bigger the font is, and the more striking the

color is. Figure 12 visually highlights the roads that played a crucial role in the road networks in different periods. Among the roads, Renmin Street, Jilin Road, and Puyang Street (western expressway) were the critical roads in 10 historical periods. Xi'an Road, Ziyou Road, and East Ring Road were the critical roads in 9 periods. The integration examines the standardized distance from any road to all other roads in the road networks, showing a trend of gradual expansion from the middle to the surrounding areas. The locations of the core regions in different historical periods were given by taking the 2017 road network as the template. As shown in Figure 13, the location of the core in the road networks constantly changed over time. However, the core positions of different eras overlapped, so the whole evolutionary cycle was divided into the former (1912–1983) and the latter (1992–2017) periods, which are illustrated in Figure 13a,b. The two figures show that the core area evolved gradually from the old city to the west and then to the south. Up to now, the core area remained around Nanhu Park. During the entire process, only in 1983 did the core area return to the old city.

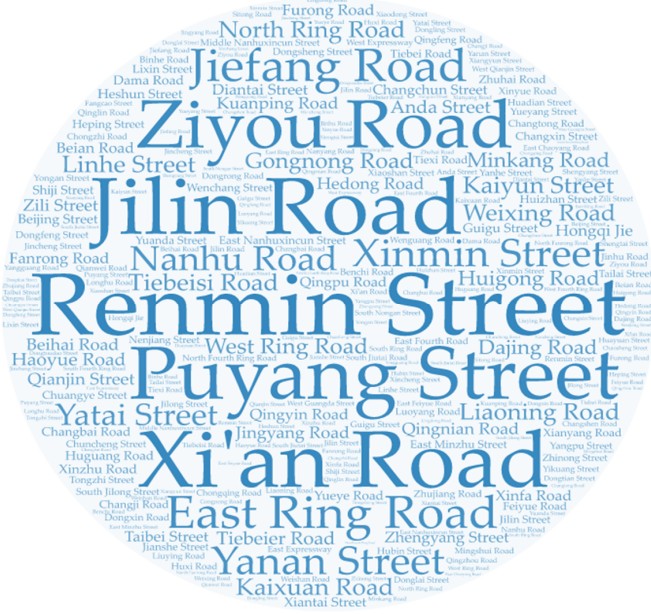

**Figure 12.** Changchun arterial roads over time (1939–2017). All names here are the key arterial roads, and their sizes show the frequency. The higher the frequency is, the bigger the font with the same color is.

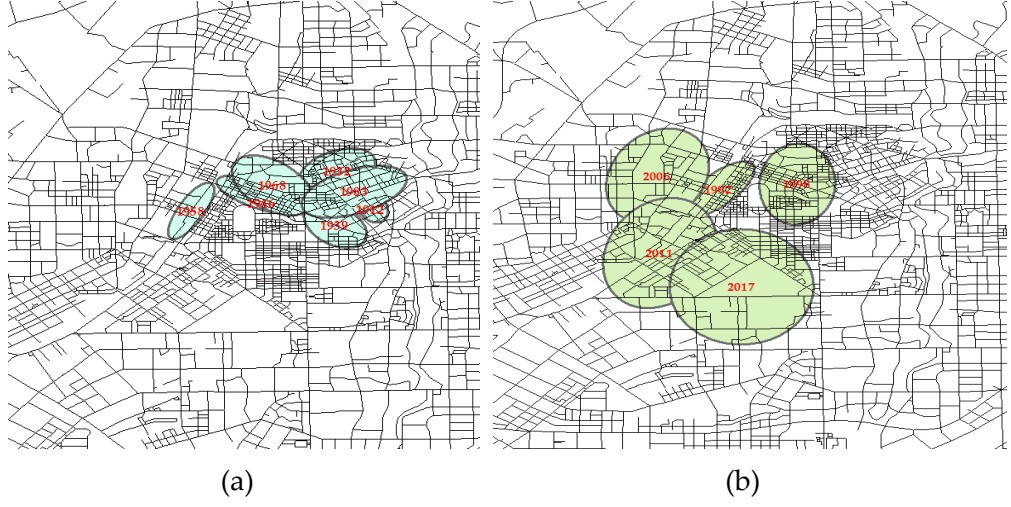

| (a) | (b) |

**Figure 13.** Core area evolution of road networks in Changchun. (**a**) 1912–1983; (**b**) 1992–2017.

*5.3. Multi-Index Evolutionary Correlation*

Besides the statistical description for the road network evolution, a more important task of the study was to explore the relationships between structure and function of the road networks. In this section, we will analyze the evolution correlation of different measures from the perspective of three scales (i.e., 1912–2017, 1932–1958, and 1968–2017) and considering the differences of the historical stages.

Table 6 shows the Pearson correlation coefficients between any two measures in the whole investigation period (1912–2017). As defined in Section 4.2, the measures were divided into two categories: state and function measures. The results exhibited that the correlation between them was significant. $\alpha$, $\beta$, and $\gamma$ are all classic measures of connectivity and degree. They had a positive, linear correlation with the degree, and the correlation coefficients were greater than 0.99. So, the degree was selected as an essential connectivity indicator for the next correlation analysis. The edges, road lengths, and number of nodes can be regarded as measures indicating the size or scale of the road networks. During the whole evolution period, network size was a crucial factor to influence other structural and functional evolutions, and the three indicators were well correlated with the average path length (+), graph density (−), average cluster coefficient (−), modularity (+), weighted path length (+), efficiency (−), and fractal dimension (+). The plus sign represents a positive correlation, and the minus sign represents a negative correlation. The correlation and significance of the three indicators with other indicators were different; the values can be found in Table 6. More important findings were based on the link between the structure and function. Results showed that average path length had a significant correlation with network size or scale (+), graph density (−), cluster coefficient (−), modularity (+), weighted path length (+), efficiency (−), and fractal dimension (+). Modularity was well correlated with network size or scale (+), average path length (+), graph density (−), average cluster coefficient (−), weighted path length (+), and efficiency (−). The weighted path length was related to network size or scale (+), average path length (+), graph density (−), modularity (+), and efficiency (−). Efficiency had a linear correlation with network size or scale (−), average path length (−), graph density (+), modularity (−), weighted path length (−), and density (+).

However, the correlation varied over time, especially under different planning styles. Tables 7 and 8 show the correlations of a variety of measures in Japanese and Chinese planning. During Japanese planning (1932–1958), the network size or scale no longer correlated with average cluster coefficient, efficiency, and fractal dimension. The significance tests of the correlation between the number of nodes or edges and weighted path length failed. In this period, the average path length was not related significantly to the fractal dimension of the urban road networks. Average cluster coefficient, weighted path length, and efficiency failed in the significance tests with average path length. Modularity also lost correlation with the average cluster coefficient and did not pass the significance test with efficiency. As for the weighted path length, it failed in the significance tests with the number of nodes or edges, path length, and efficiency, although the Pearson correlation coefficients were higher than 0.90. Similarly, efficiency was well correlated with density (+), and density was the only indicator that passed the significance test, although the correlation coefficients were higher than 0.70.

**Table 6.** Correlation among different measures (1912–2017). ** Significant at the 0.01 level; * significant at the 0.05 level. The same below.

| Pearson Correlation Coefficient | Nodes | Edges | Total Length | Degree | Path Length | Graph Density | Cluster Coefficient | Modularity | Segment Length | Weighted Path Length | Efficiency | Fractal Dimension | Density |
|---|---|---|---|---|---|---|---|---|---|---|---|---|---|
| Nodes | 1 | 0.999** | 0.921** | 0.279 | 0.942** | −0.661* | −0.669* | 0.841** | −0.414 | 0.760** | −0.618* | 0.718** | −0.211 |
| Edges | | 1 | 0.916** | 0.307 | 0.946** | −0.664* | −0.681* | 0.840** | −0.423 | 0.753** | −0.620* | 0.731** | −0.210 |
| Total Length | | | 1 | 0.183 | 0.813** | −0.604* | −0.628* | 0.774** | −0.304 | 0.903** | −0.678* | 0.596* | −0.334 |
| Degree | | | | 1 | 0.480 | −0.635* | −0.770** | 0.477 | −0.310 | 0.205 | −0.400 | 0.476 | 0.002 |
| Path length | | | | | 1 | −0.822** | −0.753** | 0.937** | −0.501 | 0.698* | −0.606* | 0.712** | −0.079 |
| Graph density | | | | | | 1 | 0.798** | −0.938** | 0.123 | −0.659* | 0.584* | −0.520 | 0.032 |
| Cluster Coefficient | | | | | | | 1 | −0.750** | 0.471 | −0.547 | 0.514 | −0.720** | 0.037 |
| Modularity | | | | | | | | 1 | −0.430 | 0.775** | −0.665* | 0.548 | −0.127 |
| Segment length | | | | | | | | | 1 | −0.019 | −0.185 | −0.351 | −0.592* |
| Weighted path length | | | | | | | | | | 1 | −0.862** | 0.370 | −0.566 |
| Efficiency | | | | | | | | | | | 1 | −0.270 | 0.789** |
| Fractal dimension | | | | | | | | | | | | 1 | 0.084 |
| Density | | | | | | | | | | | | | 1 |

**Table 7.** Correlation among measures (1932–1958). The bold represents the changed parameters compared with the whole period. The same below.

| Pearson Correlation Coefficient | Nodes | Edges | Total Length | Degree | Path Length | Graph Density | Cluster Coefficient | Modularity | Weighted Degree | Segment Length | Weighted Path Length | Efficiency | Density |
|---|---|---|---|---|---|---|---|---|---|---|---|---|---|
| Nodes | 1 | 0.996** | 0.988* | −0.101 | 0.992** | −0.973* | 0.262 | 0.981* | 0.611 | 0.666 | 0.933 | −0.730 | −0.673 |
| Edges | | 1 | 0.986* | −0.020 | 0.999** | −0.978* | 0.181 | 0.966* | 0.635 | 0.680 | 0.937 | −0.756 | −0.701 |
| Total Length | | | 1 | −0.033 | 0.987* | −0.995** | 0.218 | 0.987* | 0.722 | 0.772 | 0.977* | −0.819 | −0.772 |
| Degree | | | | 1 | 0.022 | −0.056 | −0.973* | −0.186 | 0.444 | 0.324 | 0.091 | −0.425 | −0.463 |
| Path length | | | | | 1 | −0.982* | 0.142 | 0.961* | 0.662 | 0.702 | 0.945 | −0.781 | −0.728 |
| Graph density | | | | | | 1 | −0.136 | −0.971* | −0.778 | −0.818 | −0.989* | 0.868 | 0.826 |
| Cluster Coefficient | | | | | | | 1 | 0.369 | −0.225 | −0.097 | 0.118 | 0.211 | 0.248 |
| Modularity | | | | | | | | 1 | 0.661 | 0.730 | 0.953* | −0.753 | −0.703 |
| Weighed degree | | | | | | | | | 1 | 0.991** | 0.852 | −0.983* | −0.993** |
| Segment length | | | | | | | | | | 1 | 0.890 | −0.978* | −0.983* |
| Weighted path length | | | | | | | | | | | 1 | −0.917 | −0.885 |
| Efficiency | | | | | | | | | | | | 1 | 0.997** |
| Density | | | | | | | | | | | | | 1 |

**Table 8.** Correlation among different measures (1968–2017). The underline represents the changed parameters compared with the first period.

| Pearson Correlation Coefficient | Nodes | Edges | Total Length | Diameter | Path Length | Graph Density | Cluster Coefficient | Modularity | Weighted Degree | Segment Length | Weighted Path Length | Efficiency | Fractal Dimension | Density |
|---|---|---|---|---|---|---|---|---|---|---|---|---|---|---|
| Nodes | 1 | 0.999** | 0.871* | 0.968** | 0.977** | −0.803* | −0.904** | 0.868* | −0.728 | −0.752 | 0.561 | −0.719 | 0.875** | 0.956** |
| Edges | | 1 | 0.864* | 0.971** | 0.980** | −0.801* | −0.899** | 0.866* | −0.736 | −0.759* | 0.546 | −0.711 | 0.878** | 0.958** |
| Total Length | | | 1 | 0.832* | 0.797* | −0.725 | −0.841* | 0.779* | −0.386 | −0.680 | 0.876** | −0.915** | 0.696 | 0.800* |
| Diameter | | | | 1 | 0.975** | −0.883** | −0.870* | 0.926** | −0.792* | −0.797* | 0.561 | −0.767* | 0.858* | 0.938** |
| Path length | | | | | 1 | −0.863* | −0.899** | 0.913** | −0.829* | −0.810* | 0.475 | −0.669 | 0.858* | 0.977** |
| Graph density | | | | | | 1 | 0.863* | −0.993** | 0.813* | 0.848* | −0.595 | 0.786* | −0.601 | −0.862* |
| Cluster Coefficient | | | | | | | 1 | −0.904** | 0.744 | 0.894** | −0.589 | 0.719 | −0.596 | −0.962** |
| Modularity | | | | | | | | 1 | −0.822* | −0.861* | 0.607 | −0.795* | 0.670 | 0.911** |
| Weighted degree | | | | | | | | | 1 | 0.784* | −0.056 | 0.339 | −0.570 | −0.833* |
| Segment length | | | | | | | | | | 1 | −0.442 | 0.629 | −0.422 | −0.888** |
| Weighted path length | | | | | | | | | | | 1 | −0.945** | 0.410 | 0.474 |
| Efficiency | | | | | | | | | | | | 1 | −0.560 | −0.653 |
| Fractal dimension | | | | | | | | | | | | | 1 | 0.742 |
| Density | | | | | | | | | | | | | | 1 |

The phenomenon of higher correlation coefficients and no significance may be the result of a small sample size. During the planning period (1968–2017) of the People's Republic of China, the difference between the whole period (1912–2017) and Chinese planning period is that the number of nodes or edges did not pass the significance tests with weighted path length and efficiency, and road total length failed in the significance test with fractal dimensions. Meanwhile, network size or scale showed a significant, linear correlation with density. Notably, the network diameter exhibited a significant correlation with all indicators except the weighted path length in the period, while it was related to weighted path length and not related to efficiency and density in the whole period. The correlation of the average path length with other measures was almost the same as the whole period, but it lost the correlation with weighted path length or efficiency and became correlated with density (+). Modularity in the period became well related with average segment length (−) or density (+) and lost the correlation with weighted path length. The weighted path length did not pass significance tests with the number of nodes or edges, average path length, graph density, and modularity. Efficiency failed in the significance tests with the number of nodes or edges, average path length, and density. The weighted degree was well related to network diameter (−), average path length (−), graph density (+), modularity (−), and density (−). In this period, density had a significant correlation with other measures except for weighted path length, efficiency, and fractal dimension. Between the second period (1968–2017) and the first period (1932–1958), the differences mainly were in the increased correlation of the average cluster coefficient, weighted degree, average segment length, efficiency, and density with other measures.

## 6. Discussion and Conclusions

This paper proposed a study framework based on network theory to analyze the temporal evolution of urban road networks and investigated the road network evolution of a middle-sized developing city, Changchun, China, with an almost complete historical period (1912–2017). The method integrated different disciplines and measures (network science, geographic information science, and transport science) to characterize these patterns. The correlation between structure and physical functions of road networks was determined by the road network evolution empirical analysis. The proposed method could be applied to the road network analysis of other cities, and the research results provide a series of crucial empirical evidence for the evolution pattern of the urban road networks, the link between structure and function, and road network planning and design.

If we attempt to apply our study to the urban road network planning and design, first of all, we should determine the current historical period and planning modes. Is the city road network in a stage of densification or sprawl or a combination of the two? After that, we may try to improve network performance by adjusting the structural measures of the current road network based on our findings. For example, the increase of density (the ratio of road length and real land area) in the stage of densification will improve the network efficiency and reduce the average travel distance within any OD pair. However, during sprawl or combination periods, the increase of density tends to increase the average travel distance and reduces network efficiency. Note that the travel distance here is the average distance between any two intersections of the road networks. This inspires us to construct or buy houses in a reasonable location. Housing on the outskirts of the city will significantly increase the distance of residents' travel. For the two periods, transportation engineers and management departments need to pay more attention to traffic control and travel information services to improve travel convenience in a denser area because of the increase in average path length. Meanwhile, we could better improve the connectivity of urban road networks, but not form a strong community structure to have a shorter path length. Our findings on the characteristics of road network evolution could characterize the periods of road development, based on which to optimize the road network performance, such as improving network efficiency and reducing the average travel distance. It can not only optimize the road structure in each road development stage but also provide technical support for planners and designers to build sustainable urban and transport systems.

Consistent with the contributions proposed in the literature review section, the main conclusions are centered on the following three points: network size or scale, network measure evolution, and multi-index evolution correlation.

(1)  As for the first aspect, the evolution of road networks exhibited two distinct characteristics—sprawl and densification—and the evolution patterns depended on the historical periods and urban planning modes. From the perspective of the different measures, a significant correlation existed among the number of intersections, the number of road segments, and total road length. The population was well correlated with the total road length and car ownership.

(2)  As for the second aspect, each index also presented specific rules. The number of intersections with different types in a variety of years had a significant correlation indicating a steady evolution style of the intersections, and T-junction was the most common intersection in different periods. The variation range of several indicators was broad. All road networks were small-world networks. The arterial roads have been consistent over time; however, the core areas varied in the adjacent range but were generally far from the old city.

(3)  As for the third aspect, we found correlation between structure and function of the urban road networks from the perspective of temporal evolution. The correlation changes were also related to the historical periods and planning modes. The Japanese planning period (1932–1958) could be regarded as a process of road network densification within the outer ring road, while the Chinese planning period (1968–2017) was a combination of densification and sprawl. Compared to the first period, the correlation of cluster coefficient, segment length, efficiency, and density with other indicators increased in the second period.

This study outlines a detailed, quantitative review of the road network evolution in an almost complete historical period of Changchun city. This not only makes up for the lack of existing literature but also provides a new perspective for sorting out the development history of the old city and constructing a road network database spanning a hundred years for the city. We could apply the analysis method proposed in the study to investigate the evolution pattern of road networks in other cities. However, results from the Changchun analysis are just a reference for a medium-sized developing city, and the sample size was less because of the limited urban planning period. The research conclusions still need to be validated in more empirical cases. Meanwhile, the function of the road networks in our study was measured by physical functions, and more traffic flow data may be considered to study the link between structure and function of the road networks in future research [43] as well as the coevolution of urban multilayer networks [69].

**Author Contributions:** Conceptualization, S.W. and H.Z.; Data curation, Y.L. and H.M.; Formal analysis, S.W. and H.M.; Funding acquisition, D.Y. and H.Z.; Investigation, D.Y., M.-P.K. and H.Z.; Methodology, S.W.; Project administration, D.Y.; Supervision, D.Y. and M.-P.K.; Visualization, Y.L. and H.M.; Writing—original draft, S.W.; Writing—review & editing, M.-P.K., H.Z. and Y.L.

**Funding:** This research was funded by the National Natural Science Foundation of China (Grant Nos. 51408257, 51308249), Jilin Special Fund for Industrial Innovation (2019C024), Jilin Science and Technology Development Project (20190101023JH), Jilin Education Department "13th five-year" Science and Technology Project (JJKH20180153KJ), and China Scholarship Council (Grant No.201806170188).

**Acknowledgments:** We thank Wei Yang at the Changchun Institute of Urban Planning and Design for her help during data collection, Prof. Jue Wang at University of Toronto and Lirong Kou at UIUC for GIS technique support.

**Conflicts of Interest:** The authors declare no conflicts of interest.

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
