# Peer review of "The Evolution and Growth Patterns of the Road Network in a Medium-Sized Developing City: A Historical Investigation of Changchun, China, from 1912 to 2017"

_sustainability, doi:10.3390/su11195307_

Round 1
Reviewer 1 Report
Very interesting research, with reference to the city's history. Important research in the context of contemporary transport problems in cities. The method of analyzing changes in the transport network by compacting the road network and its expansion as the city area increases is presented in a legible and understandable way.
The presented research lacks many factors that affect changes in the road network: population, sources and purposes of traffic, traffic structure - the share of various types of travel, the volume of this traffic. There is no reference to the development of public transport, bicycle traffic. There are no analyzes regarding time losses in individual means of transport.
Without such analysis, the presented research is an interesting, but only very theoretical aspect of the development of the road network. I recommend supplementing the analyzed data with the population and its change in years, the number of registered vehicles and its change in years. It would be valuable to present development of the public transport line (with types of public transport) against the background of the road network development.
Does increasing the density of the road network also improve traffic efficiency? Please try to answer this question based on your research.
Reviewer 2 Report
In this paper, the authors proposed a study framework based on network theory and investigated the road network evolution of a middle-sized city.
In my opinion the paper makes an interesting contribution. The results appear to be substantially correct, and the literature review is adequate. Approaches to results are very understandable and the methodology seems sound.
However, the expositions on topology and parameters of space syntax are deficient (Page 13-14).
Furthermore, I do not know why the authors tried to include word cloud of road names (Page 16-17).
The recommendation for the paper is to accept it after a minor amendment.
Reviewer 3 Report
This manuscript deals with the evolution and growth patterns of the road network in a medium-sized developing city of Changchun, China.
Manuscript needs a lot of improvement.
Literature Review:
Use one pattern of referencing …use number method only…. You cannot use..APA and number both.
Usually from a manuscript, three types of contribution are expected.
a-Methodological Contribution
b-Case Study-A new case
c- Addition of new variables or analysis
Which type of contribution we can have from this manuscript.??? [Please Explain]
Data collection
-How we can differentiate between Large, Medium or small city…no explanation….[Please Explain]
-Title shows you want to discuss medium city…..In this section you have mentioned
Changchun is one of the major cities in northeast China…….[Please Explain]
-Changchun is a young city developed in modern times with a history of only over 200 years. In 1800, Changchun Hall was formally established by the Qing Dynasty government (1616-1912).Until the end of the 19th century, Changchun was just an ordinary frontier market town, still in the urban formation period. So, the urban road networks of the 19th century were not analyzed here. In the 20th century, between 1904 and 1905, the Japanese empire had an imperialist war with Tsarist Russia on the land of northeast China to fight for the control of China’s Liaodong Peninsula and the Korean peninsula, which failed in Tsarist Russia. After the war, the treaty on matters concerning the three north eastern provinces was signed, which required the Qing government to develop Changchun, Harbin and other 16 cities in northeast China as trading ports of Japan to sell goods. The road network in this period as an example is shown in Figure 1. [Line 143-151-No References….Add references]
-[Line 159-210-No References.Add references]
-Please add a limitation of the study. [Please add]
-Discussion and Conclusion should be separate. [Please do]
-In Discussion section discuss how these three objectives are achieved… one by one. [Please Explain]
(1) The work analyzes the evolution pattern of a middle-sized developing city which has been reported rarely. Besides the evolution pattern, the study framework and methodological analysis workflow based on network theory proposed here could be applied to the evolutionary study of road networks in almost any city.
(2) The work proposes a characterization of network scale evolution and the network measure evolution from the perspective of multi-disciplinary integration, including network science, geographic information science, and transport science. The characteristics of road network evolution could help us define the period of road development. On this basis, the network measure results could help optimize road network performance, such as improving the network’s efficiency and reducing the average travel distance.
(3) The work determines whether transparent relationships between structure and function of the road networks exist. The study will provide empirical evidence on the correlation from the perspective of evolution. After we validate the correlation, the network theory will be more helpful for transport studies in the future.
Round 2
Reviewer 1 Report
Thank you to the authors for their comprehensive comment. The paper presents interesting research on the evolution of the road network in the city.
Author Response
Thanks for your comments to improve the quality of our manuscript.
Reviewer 3 Report
-'Literature Review' still have author's name and et al...Please remove completely.
-Discussion on achievement of three objectives still pending.Add point wise clear discussion, to connect objectives with results.
(1)
(2)
(3)
like that....
Regards
Author Response
Thanks for your comments to improve the quality of our manuscript.
Response 1: On the references, "author's name and et al" is a common citation style and widely used in the Journal "Sustainability". Please see
http://dx.doi.org/10.3390/su11185131 ;
https://doi.org/10.3390/su11185140 ;
https://doi.org/10.3390/su11195143 and so on. It is okay for "Sustainability".
Response 2: We have reorganized the conclusions as request, like (1), (2), (3) which is consistent with the section of the literature review.
This manuscript is a resubmission of an earlier submission. The following is a list of the peer review reports and author responses from that submission.
Round 1
Reviewer 1 Report
The revised manuscript addresses most of the points raised in my previous review. There are, however, some grammar problems that should be fixed as well as some poorly phrased sentences (for instance, line 119 "the work proposes the characteristics of" -> "the work proposes characterization of").
Reviewer 2 Report
Dear Authors,
you have not follow the the reviewers remarks for improving the quality of paper. you really need a big help for understanding statistics.